# Tumorigenesis Mechanisms Found in Hereditary Renal Cell Carcinoma: A Review

**DOI:** 10.3390/genes13112122

**Published:** 2022-11-15

**Authors:** Bradley R. Webster, Nikhil Gopal, Mark W. Ball

**Affiliations:** Center for Cancer Research, Urologic Oncology Branch, National Cancer Institute/NIH, 10 Center Drive, CRC Room 2W-5940, Bethesda, MD 20892, USA

**Keywords:** hereditary renal cell carcinoma, mechanisms of disease, genetics of renal cell carcinoma, cell biology of renal cell carcinoma

## Abstract

Renal cell carcinoma is a heterogenous cancer composed of an increasing number of unique subtypes each with their own cellular and tumor behavior. The study of hereditary renal cell carcinoma, which composes just 5% of all types of tumor cases, has allowed for the elucidation of subtype-specific tumorigenesis mechanisms that can also be applied to their sporadic counterparts. This review will focus on the major forms of hereditary renal cell carcinoma and the genetic alterations contributing to their tumorigenesis, including von Hippel Lindau syndrome, Hereditary Papillary Renal Cell Carcinoma, Succinate Dehydrogenase-Deficient Renal Cell Carcinoma, Hereditary Leiomyomatosis and Renal Cell Carcinoma, BRCA Associated Protein 1 Tumor Predisposition Syndrome, Tuberous Sclerosis, Birt–Hogg–Dubé Syndrome and Translocation RCC. The mechanisms for tumorigenesis described in this review are beginning to be exploited via the utilization of novel targets to treat renal cell carcinoma in a subtype-specific fashion.

## 1. Introduction

Renal cell carcinoma (RCC) has an incidence of ~400,000 new cases per year worldwide. Unfortunately, approximately one-third of patients present with or develop metastatic disease, with a 5-year overall survival of ~12% once metastatic disease develops, and RCC shows an overall mortality of 30–40% for all patients [1]. Risk factors for developing RCC are well described and include male gender, smoking, obesity, hypertension and chronic kidney disease [2]. Environmental risk factors have been described, including water pollution such as arsenic contamination, or exposure to industrial solvents or occupational exposures such as to trichloroethylene, herbicides, pesticides, asbestos and copper sulfate [3,4,5,6]. Additional work is needed to describe how environmental factors influence hereditary RCC. Those with a localized disease are typically treated with either partial or radical nephrectomy and, as mentioned, have better outcomes than those that present with metastatic disease. Furthermore, RCC is not a single entity but is a heterogenous cancer, now with 24 different recognized subtypes according to the 2022 World Health Organization’s (WHO) 5th edition RCC classification, which is an increase from the 16 entities described in the 2016 version. Prior classifications have focused on tumor morphology and immunohistochemistry; however, with each new edition the molecular and genetic underpinnings that cause tumorigenesis have come to the forefront when classifying each RCC subtype [7].

Unique genetic alterations drive tumorigenesis in each RCC subtype. These have been elegantly elucidated over the past several decades, affecting various areas of cell biology including oxygen sensing, cell growth pathways and, increasingly, mitochondrial biology [8]. These alterations lead to unique cellular behavior between subtypes, leading to unique growth rates, metabolism and metastatic potential. Many of these alterations were discovered by studying hereditary RCC which accounts for only 5% of all cases, but the information elucidated from hereditary RCC has been used to treat the remaining 95% of sporadic RCC [9]. Furthermore, specific alterations have provided opportunities for unique subtype treatment with individualized medicine, mostly in the metastatic setting. However, these alterations have increasingly been applied for localized disease to shrink tumors or even prevent new tumors [10]. This review will focus on the tumorigenesis of hereditary RCC, describe clinical features of the more common forms of hereditary RCC and highlight how knowing the tumorigenesis of an RCC subtype has led to new therapeutic options. 

## 2. Von Hippel–Lindau Syndrome (VHL)

### 2.1. Clinical Aspects of VHL

Von Hippel–Lindau Syndrome (VHL) was originally described as an autosomal dominant condition in the early 1900s by Eugene von Hippel and Arvid Lindau independently and is the most common hereditary RCC syndrome, affecting ~1/36,000 individuals [11,12,13]. However, its molecular tumorigenesis mechanisms were not described until the 1990s and beyond when the *VHL* gene, a tumor suppressor gene, was localized to chromosome 3 (3p25-3p26) [14]. The majority of VHL patients present with germline mutations; however, ~20% have sporadic de novo mutations, and mutations to *VHL* are common to sporadic clear cell RCC (ccRCC) with hundreds of different chromosome 3p mutations being described [15,16]. In fact, ~70% of sporadic RCC lesions show *VHL* mutations but with a higher degree of tumor heterogeneity due to additional lost tumor suppressor genes on chromosome 3p [17,18,19]. Additionally, of note, translocation events can also lead to VHL [20]. Clinically, VHL lesions present in several organ systems and can be benign or malignant. Hemangioblastomas may present in the liver, brain, spine and retina and may require surgical resection. VHL patients are at increased risk of pheochromocytomas, paragangliomas and renal cell carcinoma (usually clear cell RCC). Additional lesions include endolymphatic sac tumors of the inner ear, pancreatic neuroendocrine tumors that frequently need surgical resection, and cystadenomas of the epididymis of males or broad ligament of females leading to possible fertility issues despite their benign nature. Regarding renal lesions in VHL, kidney may develop thousands of microscopic or large lesions that are solid or cystic with each lesion having malignant potential. The average age for an initial lesion is the 4th decade of life, but many showing lesions much earlier [21,22,23,23]. However, one study described a cohort where the median age of onset for an initial lesion was age 28, with a median life expectancy of 66. In addition, they reported that those presenting with an initial CNS lesion correlated to worse overall survival compared to those presenting with abdominal lesions initially [24]. Numerically, microscopic examination of VHL kidneys have revealed an average of 1,000 cysts with clear cell lining (pre-malignant) and over 600 clear cell neoplasms being present by a median age of 37 correlating with the likely eventual development of larger uni-, bi- or multi-focal RCC [25]. Another, less appreciated effect of VHL includes the possibility for increased total body visceral adipose tissue deposition [26].

Lifetime penetrance for having at least one VHL-related lesion approaches 100% [27]. Specifically, regarding renal lesions, VHL has the highest penetrance for RCC amongst hereditary RCC condition, with lifetime risk being ~70% [28]. Regarding penetrance of pheochromocytomas, ~30% of VHL patients develop pheochromocytomas while only ~10% develop metastatic disease [29,30]. 

Given the wide variety of penetrations for each possible VHL-related lesion, attempts have been made to link clinical phenotype to the type of germline mutation. Type 1 VHL mutations are at a lower risk of pheochromocytoma and tend to have truncating mutations to *VHL*, while types 2A, 2B and 2C arise from missense mutations and have a higher risk of pheochromocytoma. In fact, type 2C-specific mutations only increase the risk of pheochromocytomas while hemangioblastoma risk is higher in type 1, 2A and 2B. RCC risk is highest in the missense mutations or truncating mutations of type 2B and 1, respectively [17,27,31,32,33,34]. 

### 2.2. Mechanisms of Tumorigenesis of VHL

As mentioned, VHL is acquired, like all other forms of hereditary RCC, in an autosomal-dominant fashion and follows Knudson’s ‘two-hit’ hypothesis. That is to say, the patient is born with the first hit as a germline mutation of VHL or complete loss of 3p and the second hit is acquired somatically, leading to the characteristic VHL lesion(s) over time [22]. Other mutations or alterations to chromosome 3p affecting the tumor suppressors *PBRM1, BAP1, SETD2* also occur, further promoting tumorigenesis [19]. In addition, *VHL* mutations can occur in a mosaic fashion [35]. Frequently VHL patients also acquire 5q and 8q amplifications along with 9p deletions [36]. Similar genetic changes are seen in sporadic ccRCC, however, tumor heterogeneity is generally less diverse in VHL and lesions develop later in life in the setting of sporadic ccRCC [19].

With the discovery of VHL, the biology of the tumorigenesis of the syndrome was further evaluated, with implications for potential management of the not only VHL but also other forms of hereditary RCC and sporadic tumors in both localized and metastatic disease. At the cell biology level, *VHL* encodes for an E3 ubiquitin ligase and functions with additional proteins, including Elongin B and C, cullin-2 and RBX1 [37,38,39,40]. Of note, VHL disease may be caused by mutations to VHL interacting proteins such as Elongin C, which is now its own RCC subtype according to the WHO [7,41]. Under normoxic conditions the transcription factor, hypoxia-inducible factor (HIF), is hydroxylated on proline residues by oxygen-sensing prolyl-hydroxylase enzymes (PHD). Hydroxylated HIF is then recognized by the VHL complex and covalently tagged with ubiquitin on lysine residues, marking HIF for degradation by the proteosome. During hypoxia, this degradation is attenuated through inhibition of PDH with subsequent buildup and activity of HIF. Of note, two forms of HIF exist, HIF1α or HIF2α, that form heterodimers with HIF1β/ARNT thus activating the HIF transcription network [42,43,44]. Some studies suggest that HIF2α is the major isoform driving tumorigenesis in VHL and that it therefore drove the development of an oral agent to combat VHL-related lesions targeting HIF2α (see below) [18,45]. 

The functional consequences to elevated HIF are numerous, and it is the main driver for tumorigenesis of VHL as well as sporadic and other forms of hereditary RCC. HIF acts as a transcription factor, driving cell proliferation by increasing levels of platelet derived growth factor beta (PDGFβ), transforming growth factor alpha (TGF-α) and its receptor epidermal growth factor receptor (EGFR), stimulating angiogenesis by augmenting levels of vascular endothelial growth factor (VEGF) and increasing glucose uptake and aerobic glycolysis/Warburg effect by increasing expression of glucose transporter 1 (GLUT1) [46,47]. Cell cycle control, apoptosis and motility are also altered through changes in cyclin D1, p53 and CXCR4, respectively [48]. Furthermore, VHL lesions show drastically altered metabolism in that HIF actively attenuates expression of *CPT1a*, a transporter of fatty acids into the mitochondria and the rate-limiting step of β-oxidation in the mitochondrial matrix. HIF activity therefore attenuates lipid metabolism via β-oxidation thus promoting lipid accumulation, a pathognomonic finding in ccRCC, and pushes the tumor even further towards relying on the Warburg effect through glycolysis for survival [49]. Additional Warburg effect changes, mediated by HIF, include increased expression of glycolytic enzymes (PDK, CA-9) and erythropoietin (EPO) for red blood cell production. 

Other VHL targets have been identified through non-biased screens, and future work needs to elucidate the downstream functional consequences of these additional targets with regard to cell signaling and post-translational modifications (PTMs), such as acetylation, with the hopes of identifying future treatments [50,51]. As an example, a recent additional target for VHL includes the mitochondrial transcription-promoting protein TFAM where VHL functions to stabilize TFAM, thus leading to increased mitochondrial mass and inhibition of mitochondrial proteolysis by the protease LONP1. VHL tumors are known to have low mitochondrial mass; indeed, augmenting mitochondrial mass through LONP1 inhibition may improve response to systemic agents such as the TKI, sorafenib and limit the Warburg effect [52]. The identification that mutations to both *VHL* and nearby genes on chromosome 3p can act as the first events causing tumorigenesis in several forms of RCC (both hereditary and sporadic) has opened the door to treatment approaches in a patient-specific fashion (see below).

### 2.3. VHL Therapeutic Targets and Treatment Strategies 

The treatment of VHL and its associated lesions has undergone rapid changes since the early 1990s when VHL was first described. Given the numerous potential sites for lesion occurrence, a multi-disciplinary approach is needed to treat those with VHL. This includes, but is not limited to, applications of urology, neurosurgery, surgical oncology, medical oncology, ophthalmology, otolaryngology, nephrology and endocrinology. Furthermore, active surveillance by the respective teams to detect new lesions and monitor existing lesions of the retina, brain, spinal cord, liver, adrenal and kidney is warranted through regular exams and imaging (ideally with MRI). Various guidelines exist on when to start screening, with some advocating for hearing/biochemical testing at age 5 and the obtaining of brain and abdominal MRI imaging at age 10 [53]. From a urology standpoint, lesions warranting resection from the adrenal gland or kidney are ideally resected via partial adrenalectomy or partial nephrectomy to preserve functional tissue, as most patients require multiple resections in their lifetime [23]. Partial nephrectomy of exophytic renal lesions is performed via enucleation without wide resection by utilizing the pseudocapsule plane between the tumor and normal kidney to aid in resection, thus achieving both cancer control and maintaining healthy tissue, with the goal of warding off the need for future hemodialysis [54]. More complex techniques are required for the treatment of endophytic lesions to preserve renal function [55]. By minimizing loss of normal renal parenchyma and limiting surgical ischemia time, numerous lesions may be resected in the same setting, with excellent renal functional outcomes [56,57,58,59]. Unlike those with sporadic RCC, VHL patients frequently need multiple surgeries on the same kidney in their lifetime. This places such patients at increased risk of surgical complications such as urine leaking, need for hemodialysis or bowel injury, but such re-operative procedures can be safely performed either robotically or via an open approach [60,61].

As mentioned, various lifelong active surveillance protocols have been proposed by various organizations. The VHL Alliance proposes, in contrast to others, to begin screening for lesions starting at age 16, utilizing MRI (instead of CT) to avoid chronic lifetime exposure to radiation. Imaging should be performed every 2 years if no lesions are found. Lesions in the kidney can be followed up to 3 cm following the ‘3 cm rule’ as lesions smaller than this show low metastatic potential. In fact, lesions smaller than 3 cm have a ~0% rate of spread compared to larger lesions, and their detection can be followed with more frequent abdominal imaging until they reach the full 3 cm [62]. However, lesions near the hilum or in a solitary kidney may benefit from earlier resection before the 3 cm mark is reached. Growth rates for less aggressive lesions tend to be less than 5 mm/year [23]. Growth rates, specifically for VHL lesions, are estimated on average to be 0.37 cm/year, which is faster than *FLCN* or *MET* mutation lesions, but slower than the faster-growing lesions found with *Bap1* mutation [63]. Of note, ongoing work with machine learning and MRI is attempting to develop techniques to predict individual tumor growth rates based on MRI parameters [64]. Once a lesion reaches 3 cm, it is customarily resected via the aforementioned enucleation and ideally all lesions (including accessible cysts) are resected on the ipsilateral side at that time irrespective of size to minimize the number of surgeries performed over the patient’s lifetime.

Active surveillance protocols for adrenal lesions have also been proposed, with the goal of preserving adrenal tissue while obtaining cancer control and preventing the need for adrenal hormone supplementation. Unlike the linear growth curves seen with renal lesions in VHL, adrenal growth curves behave exponentially, growing at 0.03 cm/year when < 1cm in size but increasing to greater than 0.3 cm/year once larger than 2 cm. In addition, lesions greater than 3 cm in size are less likely to be amenable to partial adrenalectomy. In addition, catecholamine release and symptoms tend to worsen with size. For these reasons, a ‘2 cm rule’ has been proposed to maximize cancer control, minimize symptoms and allow for partial adrenalectomy [33,34]. Active surveillance protocols require periodic imaging by abdominal MRI and/or [68Ga]DOTANOC PET or [131I]MIBG along with plasma-free metanephrines or 24 h catecholamine collection [65].

Treatment for metastatic lesions has targeted the known changes induced by *VHL* loss including using TKIs, such as sunitinib, to target the receptors upregulated with HIF, accumulation including VEGFR, PDGF and EGFR. Current approaches have incorporated the use of immunotherapy (I/O) therapy and TKIs either individually or in combination for metastatic ccRCC [66,67,68].

More recently, direct inhibitors to HIF2α have been developed for the treatment of localized renal and CNS tumors as well as pancreatic lesions in VHL patients. The HIF2α inhibitor, belzutifan, is approved for oral daily use with the goal of slowing tumor growth, thus offering a medical solution over surgery. A recent phase II study using belzutifan showed an ORR of 49% and stable disease in another 49% of patients. Meanwhile PFS was over 95% at 2 years while side effects were minimal, with fatigue and anemia being the most common [45,69,70]. In addition, responses were seen for 77% of pancreatic lesions, 30% of CNS hemangioblastomas and 100% of retinal lesions [69]. Overall, use of daily HIF2α inhibitors will likely reduce the need for surgical interventions in select VHL patients. Based on these results, Belzutifan was approved by the FDA in August 2021 for use in VHL patients with CNS, RCC and pancreatic neuroendocrine lesions that do not require immediate surgery [71]. Belzutifan’s use in sporadic RCC and other hereditary forms of RCC (such as HLRCC with augmented HIF activity) is also being considered.

## 3. Hereditary Papillary Renal Cell Carcinoma (HPRC)

### 3.1. Clinical Aspects of HPRC

Germline mutations to the proto-oncogene *MET* (chromosome 7q31) leads to hereditary papillary renal cell carcinoma (HPRC) forming. These *MET* mutations lead to constitutional activation of MET, which is a tyrosine kinase receptor localized on the cell surface of renal epithelial cells and normally functions in part in renal tubular repair following ischemia, chemical insult or renal hypertrophy [72,73]. Interestingly, *MET* mutations or alterations can be found in other cancers including hepatocellular carcinoma, endometrial, breast, gastric and squamous cell carcinoma of the head and neck [74]. Originally discovered in 1994 as an autosomal dominant condition within a family presenting with RCC over several generations, HPRC is unique amongst hereditary RCC in that the only phenotype seen is increased risk of bilateral or multi-focal RCC [75,76]. For those with known or likely pathogenic *MET* mutations causing HPRC, the risk of developing bilateral and/or multi-focal RCC is >80%, with some reporting penetrance rates of RCC to be >90%, which is amongst the highest rates of penetrance seen in hereditary RCC, with a median age of renal tumor diagnosis being 57 [77,78]. In addition to causing HPRC, *MET* missense mutations have been reported in ~13% of sporadic papillary RCC, and such mutations have been cor related to higher stage and possible worse overall survival [79]. Despite only 13% of sporadic type I papillary RCC tumors showing mutations in *MET*, alterations to *MET* are seen in 81% of sporadic type 1 papillary tumors due to a combination of missense mutations, gene fusion events and multiple copies of chromosome 7 (along with chromosomes 16, 17 and 20), thus making mutations to *MET* far less common amongst sporadic type 1 papillary RCC patients [80,81]. Of note, MET copy number increases cellular dedifferentiation, lymph node invasion and risk of metastasis [82]. Clinically, the tumors seen in HPRC patients behave more indolently than those from VHL patients, in that growth rates are ~0.15 cm/year compared to 0.37 cm/year in VHL tumors [63]. In addition, these type 1 papillary tumors appear hypo-vascularly on CT/MRI when compared to other RCC subtypes, such as clear cell RCC [83].

### 3.2. Mechanisms of Tumorigenesis of HPRC

*MET* mutations lead to the auto-dimerization, autophosphorylation and constitutive activation of its cytoplasmic tyrosine kinase (TK) domain (amino acids 1110 to 1268), independent of its only known endogenous ligand hepatocyte growth factor (HGF). This is the signature event in HPRC’s tumorigenesis [73]. MET functions upstream of numerous signaling pathways promoting tumorigenesis including PI3K/AKT/mTOR, STAT, MEK/ERK, GRB2, GAB1 and RAC1 thus promoting cell survival, proliferation, angiogenesis and mobility and being similar to the tumorigenesis found in certain forms of hepatocellular carcinoma [84]. Additionally, MET and VHL crosstalk may occur, as MET may regulate VEGF signaling and downstream angiogenesis even after VEGFR inhibition. This suggests that breakthrough resistance mechanisms develop after treatment with VEGF inhibitors, such as bevacizumab. Meanwhile, MET expression levels can be altered in clear cell RCC after VHL mutation [73,85,86]. This constitutive activation of a TK receptor is the basis for targeted therapies in HPRC utilizing various TK inhibitors, and likely explains the differential response to these drugs in HPRC (missense mutations) vs. sporadic type 1 papillary RCC (gene fusion and aneuploidy events) (see below). An additional mechanism for tumorigenesis following *MET* alteration includes changes to cell contact inhibition, motility and epithelial to mesenchymal transition (EMT) due to decreased E-cadherin and increased N-cadherin/vimentin expression [73,87].

### 3.3. HPRC Therapeutic Targets and Treatment Strategies

Given that the kidney of an HPRC patient may contain over 3000 microscopic tumors with malignant potential, lifelong surveillance with abdominal imaging is imperative in the treatment paradigm of HRPC [88]. The preferred lifelong surveillance regimen includes cross-sectional abdominal imaging with CT or MRI scans every 1–2 years starting at age 30, as described by NCCN guidelines [67]. The age of 30 is typically later than other forms of hereditary RCC due to the later onset of presentation in HPRC of renal lesions, but early-onset HPRC has been described [77]. Given the indolent behavior of tumors and similar to instances of treatment with VHL-related lesions, HPRC lesions can be assessed following the ‘3 cm rule’, in that these lesions do not require resection until reaching 3 cm in diameter due to the low metastatic potential of smaller lesions. Once a lesion reaches 3 cm in size, the preferred resection method is enucleation of the lesion via partial nephrectomy [89,90].

Currently, there is no approved treatment specifically for HPRC lesions, but several studies have hinted at targeting MET due to its central link in the tumorigenesis of HRPC, particularly in the context of those patients with activating mutations to *MET* (as opposed those with aneuploidy and gene fusion alterations). MET-targeting agents such as foretinib, crizotinib, savolitinib and cabozantinib, have resulted in varying levels of success [91,92,93,94,95]. As part of a phase II trial, foratenib with activity against MET, VEGFR2 and AXL showed modest benefit against type 1 papillary RCC in 74 patients with known germline or somatic mutations to MET, MET amplification or gain in chromosome 7. ORR was only 13.5%, with an MFS of 9.3 months. However, those with germline MET mutations showed a much higher response rate of 50%, suggesting that those with *MET* mutations may benefit most from this targeted therapy compared to patients with aneuploidy [94]. In a phase III study (SAVOIR) comparing sunitinib (a TKI without activity against MET) vs. savolitinib (TKI against MET), savolitinib had a numerically higher ORR, OS and PFS than sunitinib (median PFS was not statistically significant), and savolitinib had fewer grade 3 or higher adverse events and was better targeted. Of note, the majority of patients in SAVOIR had known gain in chromosome 7 alterations, and it is unknown if those with germline mutations to *MET* would respond at higher rates after treatment with savolitinib rather than sunitinib [92]. Additional studies are ongoing to investigate the efficacy of sunitinib, cabozantinib, crizotinib and savolitinib) and capmatinib/INC280 (NCT02761057, NCT02019693).

The use of checkpoint inhibitors/immunotherapy (IO) in clear cell RCC (ccRCC) is now the standard method of care. However, their use in non-ccRCC is less well studied. One study showed that ~32% of type 1 papillary RCC tumors do in fact express PD-L1 suggesting a role of IO therapy [96]. Using PD-L1-targeting pembrolizumab in papillary RCC showed an ORR of over 25%, while using CTLA-4 targeting durvalumab with savolitinib had the ORR go to over 27% [97,98]. As in the aforementioned TKI studies, better stratification is needed in future studies to elucidate whether IO therapy has differential benefits on those with germline *MET* mutations compared to patients with somatic, gene fusion or aneuploidy alterations. Overall, the tumorigenesis pathways that are triggered following *MET* alterations promise potential treatment options for those with HPRC due to their lifelong risk of renal tumors with potential metastatic disease.

## 4. Succinate Dehydrogenase-Deficient Renal Cell Carcinoma (SDH-RCC)

### 4.1. Clinical Aspects of SDH-RCC

Like other forms of hereditary RCC, SDH-RCC is inherited in an autosomal dominant fashion as a germline mutation to the succinate dehydrogenase complex, as initially described in 2004 before beingg officially recognized as a distinct form of RCC by the WHO’s RCC classification in 2016 [99,100]. Succinate dehydrogenase is a tetramer protein complex localized to the inner mitochondrial membrane composed of the subunits encoded by the *SDHA, SDHB, SDHC,* and *SDHD* genes with complex formation aided by *SDHAF1* and *SDHAF2*. Mutations to each subunit or SDHAF2 have been linked to developing SHD-RCC, with *SDHB* being the most common [101,102,103]. Recently, a patient presenting with pancreatic neuroendocrine tumors was noted to have mutations to both *SDHA* and *VHL*, but it is unknown how potential RCC tumors would behave with this dual mutation burden [104]. Succinate dehydrogenase is unique in that it functions as both a key component of the Kreb’s/Tricarboxylic acid (TCA) cycle and the electron transport chain as Complex II, thus placing the tetramer at the center of cell metabolism [105]. SDH-RCC is an overall rare form of RCC, accounting for less than 0.25% of all RCC cases and presenting at an average age of 35 years. However, it can present in the 2nd–8th decades of life [102,106] as well. Unlike HPRC, SDH-RCC patients can present with other manifestations other than RCC, including paragangliomas/pheochromocytomas (especially to the head/neck), gastrointestinal tumors (GIST) and pituitary adenomas [107,108,109]. Other studies have reported a potential increased risk of seminomas, papillary thyroid carcinoma, renal adenomas, and pancreatic neuroendocrine tumors [110]. Due to its rarity, the penetrance of each possible phenotype is unknown, however, the penetrance for RCC is estimated to be ~15% with regard to *SDHB* germline mutations [111]. Patients may develop solitary, bilateral or multifocal lesions in up to 30% of cases [112]. Pathologically, lesions are eosinophilic with variable architecture that has been described as either nested, solid or tubular. However, immunohistochemical staining for SDH loss is pathognomonic in terms of diagnosis, reflecting the instability of the tetrameric complex of SHD following the mutation of any individual subunit of the complex or *SDHAF2* [113,113,114].

### 4.2. Mechanisms of Tumorigenesis of SDH-RCC

The nuclear-encoded subunits of SDH are transported to the mitochondria, where they are assembled into the tetrameric SDH complex in the inner mitochondrial membrane. SDH is also known as succinate: ubiquinone oxidoreductase or mitochondrial complex II. SDH is a component of the TCA cycle which oxidizes succinate to fumurate, yielding the reducing equivalent FADH_2_. SDH also functions as Complex II of the electron transport chain, utilizing the aforementioned FADH_2_ to shuttle electrons to ubiquinone and Complex III and thus generating the proton gradient in the intermembrane space needed for ATP generation by Complex V/ATP synthase [102,105]. Loss of SDH expression has also been found in sporadic clear cell RCC with lost *VHL* but no *SDH* mutations, a phenomenon which was correlated with poor prognosis [115]. The link between VHL and attenuated SDH activity in the sporadic setting was explored by Aggarwal et al. who showed that HIF upregulates miR-210 following *VHL* loss which directly inhibits the *SDHD* transcript, further highlighting the tumorigenic potential of the SDH complex outside of direct *SDH* mutations [115]. Disruption of SDH disrupts the TCA cycle, thus promoting aerobic glycolysis (Warburg effect), glucose uptake and accumulation of the oncometabolite succinate. Succinate accumulation in the mitochondrial matrix is thought to inhibit proline hydroxylase (PHD) of HIF and thus the loss of VHL-mediated degradation of HIF [116]. Augmented HIF activity then promotes tumorigenesis through increased aerobic glycolysis and angiogenesis similar to the tumorigenesis of VHL [117,118]. In addition, elevated succinate after *SDH* mutation may lead to the succinylation of lysine residues as a PTM of numerous protein targets and this altered regulation of succinylation likely contributes to tumorigenesis, highlighting the role of PTMs through oncometabolites in RCC tumorigenesis [119,120,121,122,123]. Furthermore, the SDH complex mutation may inhibit cell death pathways such as the emerging ferroptosis mechanisms [124]. Further mechanisms of tumorigenesis with succinate accumulation are through the inhibition of DNA repair mechanisms and fatty acid accumulation [103,125].

### 4.3. SDH-RCC Therapeutic Targets and Treatment Strategies

Given its rarity, no clear guidelines exist for the surveillance and treatment of SDH-RCC, but most advocate for annual or bi-annual abdominal imaging, periodic neck imaging for paragangliomas, and the evaluation of plasma-free metanephrines or 24 h urine catecholamines [28]. Once a tumor has been identified, the general recommendation is for immediate surgical removal due to the aggressive nature of SDH-RCC lesions, and the ‘3 cm rule’ followed in VHL and HPRC is not recommended in SDH-RCC due to the metastatic potential of even smaller lesions well below 3cm, with metastatic risk approaching 30% [102,113,126]. During resection for smaller lesions, wide resection rather than enucleation is the general recommendation due to the aggressive nature of these lesions. In addition, for those lesions not amenable to partial nephrectomy, radical nephrectomy is performed with possible regional lymph node resection.

For metastatic lesions, ongoing studies are attempting to take advantage of knowledge gained from the study of tumorigenic mechanisms in SDH-RCC, including VEGF inhibition due to succinate-mediated HIF augmentation. This includes the use the use of the TKI cabozantinib, with activity against VEGFR2, and Nivolumab (anti-PD-1) specifically in patients with non-clear cell RCC including SDH-RCC (NCT03635892). Another ongoing study is utilizing the defective DNA repair seen in SDH-RCC by utilizing PARP inhibitors (Talzoparib) with Avelumab (anti-PD-L1) (NCT04068831). However, given the aggressiveness of these lesions, more studies are needed to identify the ideal treatment of metastatic lesions, including the role of metastasectomy in combination with systemic therapy.

## 5. Hereditary Leiomyomatosis and Renal Cell Carcinoma (HLRCC)

### 5.1. Clinical Aspects of HLRCC

In the early 1970s, several groups described families presenting with both uterine and cutaneous leiomyomas that became known Reed Syndrome [127]. The link between Reed Syndrome and the increased risk of an aggressive form of RCC was not described until the early 2000s [128]. HLRCC was initially described as type II papillary RCC, however, with the 2016 WHO classification, HLRCC is now considered a distinct entity from sporadic type II papillary RCC [99]. In fact, the overall diagnosis of papillary RCC will continue to evolve as more molecular characteristics are applied to the WHO’s classification system, further delineating papillary subtypes [7]. Pathologically, HLRCC lesions appear as high-grade aggressive and invasive lesions, have eosinophilic staining and display characteristic peri-nuclear clearing with papillary architecture that can also be solid or tubular in nature [129]. HLRCC renal masses present as unifocal, bilateral or multi-focal masses with a wide variety of sizes, with some being under 3cm while others present as over 20 cm, with metastatic potential being present for even the smaller lesions [114]. Patients frequently present with metastatic disease. Additionally, of note, these renal masses can present as cystic or solid, with malignant potential being present for all cystic lesions owing to the malignancy of the cells lining the cystic space, thus making avoidance of cyst rupture imperative to prevent tumor seeding [130]. As with other hereditary forms of RCC, HLRCC is inherited in an autosomal dominant manner and follows Knudsen’s two-hit hypothesis as both alleles must be lost before disease presentation progresses with the first ‘hit’ being the inherited germline mutation and the second being lost later in life as a loss of heterozygosity [114]. Of note, bi-allelic inheritance of *FH* mutations leads to the autosomal recessive condition fumurase deficiency at birth which leads in turn to microcephaly, dystonia, seizures and developmental delay [131].

HLRCC is caused by the germline mutation of the tumor suppressor *fumurate hydratase (FH)* at the 1q42.3-q43 locus of chromosome 1 [132]. FH, like SDH, functions as part of the TCA cycle, converting fumurate to malate with loss of FH leading to accumulation of the oncometabolite fumurate and subsequent succination of mitochondrial protein cysteine residues as a PTM (see below). Elevated cysteine succination is pathognomonic for HLRCC and is detected via increased S-(2-succinyl) cysteine IH staining, confirming the final diagnosis of HRLCC [133].

Patients with HLRCC are at increased risk of uterine fibroids, cutaneous leiomyomas and HLRCC-associated RCC, with each feature having varying penetrance and the risk of developing RCC being as high as 35% [134]. Cutaneous lesions have the highest penetration and are unique in that they are frequently painful (unlike the cutaneous lesions found in Birt–Hogg–Dubé, see below). In addition, uterine fibroids have a higher penetrance than RCC lesions and often lead to hysterectomy at an early age, prompting the need for family planning discussions with patients [135,136,137]. Renal lesions can present at various ages, but lesions tend to present at a younger age compared to sporadic RCC with ~10% of lesions presenting before age 20 [138].

### 5.2. Mechanisms of Tumorigenesis of HLRCC

Given that FH is central to the TCA cycle and subsequent mitochondrial ATP production, loss of FH attenuates the TCA cycle and oxidative phosphorylation leading to the cell’s reliance for energy on the Warburg effect and aerobic glycolysis [139]. This increased utilization of glucose is utilized clinically to detect new HLRCC primary and metastatic lesions via by fluorodeoxyglucose positron emission tomography [140].

As mentioned, loss of FH leads to accumulation of the TCA oncometabolite fumarate, similar to the buildup of succinate in SDH-RCC, and fumarate accumulation then promotes the PTM succination to numerous cysteine residues on various proteins as a mechanism to further promote tumorigenesis [141]. This highlights a recurring theme amongst hereditary RCC of mitochondrial oncometabolite accumulation which causes various mitochondrial and cellular PTMs (i.e., acetylation of lysine residues in VHL lesions; succinylation of lysine residues with SDH-RCC and succination of cysteine residues in HLRCC) and thus promotes tumorigenesis. Succination’s effects on cell metabolism can be seen through the succination of the mitochondrial DNA polymerase leading to diminished mitochondrial DNA (mtDNA) and decreased proofreading capabilities and thus more mtDNA mutations thus effectively limiting oxidative phosphorylation and mitochondrial function. The diminished mitochondrial function further promotes the Warburg effect [142]. In addition, similar to succinate inhibition of PHD in SDH-RCC, fumurate accumulation is thought to inhibit PHD leading to HIF activation with subsequent augmented VEGF and GLUT1 activities. Succination of the ubiquitin E3 ligase, KEAP1, is also seen in HLRCC leading to augmented levels of the anti-oxidant protein NRF2, thus allowing HLRCC tumors to ward off the deleterious effects from increased oxidative stress inherently found with mitochondrial dysfunction and tumorigenesis [140,141,143].

### 5.3. HLRCC Therapeutic Targets and Treatment Strategies

Management of HLRCC renal masses is similar to that of those found in SDH-RCC due to their aggressive nature, even for small lesions, and thus does not follow the ‘3 cm rule’. Therefore, all lesions should be excised with wide margins when partial nephrectomy is chosen, regardless of size. Radical nephrectomy, often via an open approach, is utilized more frequently due to the aggressive nature of lesions, and regional lymph node dissection should be considered. Cystic lesions should be excised carefully to prevent tumor spillage and seeding. In addition, aggressive yearly surveillance with MRI (with small slice size) is recommended starting at ~8 years old. Surveillance is cost effective starting at age 11 with regard to life years gained, quality-adjusted life years and net monetary benefit vs. no screening at all [144]. Furthermore, family members of those with confirmed *FH* mutations should undergo genetic screening at a young age [136,145].

Treatment of metastatic disease in HLRCC has slowly progressed as the mechanisms of tumorigenesis after *FH* loss have become known. Due to augmented HIF activity in HLRCC, Srinivasan et al. targeted the VEGF pathway with bevacizumab in combination with erlotinib, an EGFR TKI inhibitor, as part of a phase II study (NCT01130519). This study enrolled patients with advanced sporadic type II papillary RCC or advanced HLRCC associated renal lesions treated with the combination bevacizumab and erlotinib. Response rates were two times higher in the HLRCC associated lesions compared to the sporadic tumors (ORR 72% vs. 35%). In addition, median PFS in HLRCC far outpaced those of the sporadic tumors (21.1 months vs. 8.7). Based on this prospective study, bevacizumab/erlotinib treatment has been incorporated into treatment recommendations for HLRCC by the National Comprehensive Cancer Network (NCCN) [146,147]. Recent work has described, in a retrospective fashion, that checkpoint inhibitors with TKI treatment may show a more favorable response with regard to OS and PFS than bevacizumab/erlotinib [148,149,150]. Further work is needed to identify additional targets in HLRCC tumors that will be derived from a better understanding of the tumorigenesis mechanisms that arise following the loss of *FH.*

## 6. BRCA-Associated Protein 1 Tumor Predisposition Syndrome (BAP1)

### 6.1. Clinical Aspects of BAP1

As mentioned, germline mutations to chromosome 3p portends to an increased risk of hereditary RCC, and like *VHL, BAP1* mutations can lead to hereditary RCC as initially described in 2011 within a family with multi-generational RCC without known germline mutations. Family members were eventually found to have alterations to *BAP1* and not *VHL*. BAP1 syndrome increases the risk of aggressive RCC, cutaneous melanocytic lesions, basal cell carcinoma, mesothelioma (frequently in the abdomen) and uveal melanoma [151,152,153]. Furthermore, sporadic ccRCC shows mutations to *BAP1* in ~15% of cases and loss of *BAP1* in the somatic setting are associated with lower overall and cancer-specific survival [154,155,156,157]. *BAP1*-related RCC lesions tend to be aggressive, high-grade, multifocal and occur early in life. Furthermore, BAP1-related RCC has the highest known growth rates amongst hereditary RCC subtypes at 0.6 cm/year. Therefore, active surveillance is not recommended for known lesions [63]. Due to its recent discovery, the full range of the syndrome and the penetrance of each lesion is unknown and additional associated tumors may include breast, neuroendocrine, thyroid, bladder and lung [158,159]. In addition, some advocate screening and reporting BAP1 status for renal biopsies showing ccRCC in that *BAP1* mutation status can predict prognosis similar to grade and sarcomatoid/rhabdoid change [160].

### 6.2. Mechanisms of Tumorigenesis of BAP1

BAP1 functions as a tumor suppressor and deubiquitinating enzyme, regulating the cellular processes of DNA repair, genomic stability and cell cycle regulation following Knudson’s two hit hypothesis, in a manner similar to VHL [161,162,163,164]. Additionally, similar to VHL, BAP1 functions as a protein complex [165,166]. Functioning in the nucleus, BAP1 deubiquitinates targets such as HCF-1 and chromatin. BAP1 therefore regulates transcription, the cell cycle and proliferation [154]. Furthermore, BAP1 appears to function in the cytosol by regulating calcium efflux and subsequently regulates apoptosis with loss of *BAP1* inhibiting apoptosis in the face of increased DNA damage [167]. Future work is required to identify additional BAP1 targets in the hopes of finding targets to specifically treat those with *BAP1* mutations.

### 6.3. BAP1 Therapeutic Targets and Treatment Strategies

Recommendations for the clinical management of those with BAP1 syndrome have been developed, including obtaining chest and abdominal imaging every 2 years starting at age 30 [168]. Treatment and screening also require a multi-disciplinary team including treatment via dermatology, ENT, as well as general and thoracic surgery for potential uveal melanoma and chest/abdominal mesothelioma. Given the aggressive nature of and limited treatment options for metastatic disease with BAP1 RCC, the ‘3 cm rule’ is not recommended, and lesions should be resected with partial nephrectomy, and wide resection should be performed as soon as lesions are discovered [63].

## 7. Tuberous Sclerosis

### 7.1. Clinical Aspects of Tuberous Sclerosis

First described in the late 19th century, tuberous sclerosis (TSC) has an estimated incidence of 1/6000 to 1/10,000 persons worldwide [169,170]. This clinical syndrome results from germline loss of function of tumor suppressor genes *TSC1* on chromosome 9q34 (hamartin) or *TSC2* on chromosome 16p13 (tuberin) [171,172,173]. In contrast to other hereditary syndromes, 60% of germline *TSC* mutations occur as a de novo phenomenon as opposed to an autosomal dominant inheritance (Shagreen patches and periungual fibromas) [174]. A wide variety of lesions are associated with TSC, such as skin (angiofibromas, ash-leaf spots); brain (subependymal giant cell astrocytoma and cerebral cortical tubers); heart (rhabdomyoma); kidney (angiomyolipoma, cysts, and renal cell carcinoma); and lung (lymphangioleiomyomatosis or LAM). Other clinical features include epilepsy, behavioral disorders, and intellectual disabilities [175,176].

Over 80% of patients with TSC have renal manifestations, with development beginning in childhood [177]. Angiomyolipomas (AML) are by far the most common kidney lesion, occurring in 80% of TSC patients, and are often bilateral/multifocal [178]. The presence of fat on imaging is diagnostic for AML, although up to 1/3rd of lesions may be fat-poor and therefore difficult to distinguish from malignancy without a renal biopsy [176,179]. Of note, AMLs grow faster in patients with TSC as compared to the sporadic population (1.25 cm/year vs. 0.19 cm/year, respectively, with average follow-up of more than 3 years) [180].

In contrast, only a small minority (2–4%) of TSC patients develop renal cell carcinoma. Average age of onset is 30–40 years, although tumors have been seen in children as young as 7 years old [181,182,183]; additionally, tumors are female-predominant. While the most commonly associated RCC in TSC was initially thought to be ccRCC, it is now recognized that there are three subtypes distinct from traditional ccRCC: TSC-associated RCC with fibromyomatous stroma (also known as TSC associated papillary RCC or RCC with leiomyomatous features); TSC associated oncocytic tumor (referred to as HOCT by some pathologists); and eosinophilic solid and cystic tumor. Between 42–50% of TSC patients with renal malignancies present with multifocal RCCs, and a quarter of them have bilateral involvement [181,182].

### 7.2. Mechanisms of Tumorigenesis of Tuberous Sclerosis

The causative genes for TSC, *TSC1* and *TSC2* were first identified in 1997 and 1993, respectively. The resultant proteins form a complex that, through the GTPase-activating protein (GAP) domain of TSC2, catalyzes the conversion of G protein complex Rheb-GTP to Rheb-GDP [184]. This product acts to inhibit the activity of mTORC1, which normally promotes protein and lipid synthesis through the action of p70S6K and 4E-BP.; glycolysis and ATP production; lysosomal/mitochondrial biogenesis; and autophagy [185]. Thus, the TSC complex is a negative regulator of cell growth through mTORC1 inhibition. In contrast, the TSC1-TSC2 complex activates mTORC2, implicated in cell survival and cytoskeletal organization, although the underlying molecular mechanisms are not well understood [186].

Genetic alterations affecting the formation of TSC complex (in either *TSC1* or *TSC2*) result in inappropriate activation of mTORC1 and unchecked cell growth. Interestingly, although TSC is a tumor predisposition syndrome, the majority of lesions are benign (i.e., hamartomas) as opposed to malignant. Furthermore, mutations in *TSC* are not typically seen in sporadic renal tumors. This finding may be explained through concomitant loss of Akt signaling through both mTORC1 feedback and loss of mTORC2 in TSC-deficient lesions [184,186].

### 7.3. Tuberous Sclerosis Therapeutic Targets and Treatment Strategies

The identification of uncontrolled mTORC1 activation as a molecular mechanism responsible for tumorigenesis with loss of TSC complex makes mTOR inhibitors a suitable candidate for treating such tumors such as AMLs.

Traditionally, AMLs can be followed serially on imaging until they reach 4 cm, at which time intervention such as selective angioembolization [179] or nephron sparing surgery [187,188] is recommended due to increased risk of symptoms or spontaneous hemorrhage occurring at tumors beyond this size (or smaller in pregnant female patients) [189,190]. More recent evidence, however, has questioned the “4 cm approach” [191], with intralesional aneurysm size >5 mm being a more specific marker of risk of hemorrhage with conservative management [192]. When an mTOR inhibitor, everolimus, was given to TSC patients with AMLs 3 cm or larger, 42% of patients had a reduction greater than 50% in volume, with 92% progression-free survival at 12 months. The drug had an acceptable safety profile, with the most commonly reported side effect being stomatitis [193]. Growth of AMLs was noted after discontinuation of treatment, indicating that mTOR inhibition results in tumor cells being in a quiescent phase as opposed to these cells being eliminated [194]. Nevertheless, mTOR inhibitors are an option for TSC patients with AML, particularly those unfit for surgery.

Case studies have demonstrated mTOR inhibitor response to TSC associated RCCs [195,196,197]. It remains to be seen whether these targeted agents are more efficacious for these tumors as compared to PD/PDL-1 inhibitors. Regular cross-sectional imaging (MRI preferred) is recommended by national guidelines to be every 3–5 years beginning at age 12 [67,198]. More frequent imaging may be warranted depending on the size and/or growth of a suspected RCC. Surgical series demonstrate that these tumors are relatively indolent, with only 1 patient with localized RCC developing metastasis in 10 years. [181,182,183] Thus, active surveillance until the dominant lesion reaches 3 cm is applied for suspected RCC in TSC.

## 8. Birt–Hogg–Dubé Syndrome

### 8.1. Clinical Aspects of Birt–Hogg–Dubé Syndrome

In 1977, three Canadian dermatologists (Birt, Hogg, and Dubé) noted the presence of painless fibrofolliculomas, flesh colored papules originating from hair follicles in the face; neck; chest; and upper back, in certain families, suggesting a hereditary predisposition [199]. Pulmonary cysts, occasionally associated with spontaneous pneumothorax, were later found to be associated in patients with so-called Birt–Hogg–Dubé syndrome (BHD) [200]. Beginning in 1993, patients containing these fibrofolliculomas were also identified as having bilateral and/or multifocal renal tumors [201,202]. The gene responsible for the cutaneous, pulmonary, and renal manifestations in this inherited syndrome was localized in 2002 to chromosome 17p11.2 and was named folliculin (*FLCN*) [203,204].

The vast majority of BHD patients have fibrofolliculomas and pulmonary cysts; in contrast, renal tumors occur in 12–34% of patients at a median age of 50 years [205,206,207]. A range of tumor histologies can be seen, with hybrid oncocytic (combination of chromophobe and oncocytic features) being the most common (50%). Other subtypes include chromophobe (34%), clear cell (9%), oncocytoma (5%), and papillary (2%) [208].

### 8.2. Mechanism of Tumorigenesis in Birt-Hogg-Dube Syndrome

Despite its identification two decades ago, our understanding of the role of the tumor suppressor gene folliculin in promoting renal carcinogenesis is incomplete at present. Preclinical FLCN-deficient mouse models demonstrated activation of AKT-mTOR pathway, with the mTOR inhibitor rapamycin halting renal cyst and tumor growth in these animals [209,210]. However, responses in humans to mTOR inhibitors, namely for fibrofolliculomas, were not seen [211], suggesting that alternative oncogenic pathways exist beyond mTOR activation in FLCN-deficient tumors. Indeed, FLCN can either inhibit or activate mTOR depending on the particular cellular state [203]. For instance, moderate levels of transcription factors E3 and B (TFE3 and TFEB) were seen in FLCN-deficient cells. Additional investigation unearthed that loss of function in FLCN inhibits mTORC1-dependent phosphorylation of TFE3/TFEB, resulting in nuclear localization and the activation of these transcription factors, potentially contributing to oncogenesis [212,213,214]. However, despite losing its inhibiting activity on TFEB/TFE3 with *FLCN* loss, mTORC1’s activity on augmenting protein translation through activity of 4E-BP and p70S6K remained intact and was augmented even more with *FLCN* loss. In fact, knockdown of *FLCN* led to tumorigenesis in mice, but this was blocked in the dual knockout of *TFEB* and *FLCN*, thus highlighting the crosstalk between TFEB, mTOR and FLCN [215]. FLCN has also been implicated in membrane trafficking, serving as a guanine exchange factor (GEF) for Rab GTPases such as Rab7A, which is involved in endosomal recycling and lysosomal degradation of EGFR [216]. Thus, FLCN deficiency results in loss of Rab7A-mediated degradation of EGFR, resulting in inappropriate persistence of this tyrosine kinase receptor and unchecked cellular growth. Still further roles for FLCN include autophagy, ciliogenesis, cell–cell adhesions, and regulation of HIF1-α [203,217,218,219].

### 8.3. Birt–Hogg–Dubé Syndrome Therapeutic Targets and Treatment Strategies

Given our evolving understanding of the molecular mechanisms through which loss of FLCN induces tumor formation, there is at yet no reliable systemic agent for the management of the clinical phenotypes of this hereditary syndrome. Fibrofolliculomas and pulmonary cysts have no malignant potential and therefore do not require management, unless there are significant cosmetic concerns or an associated pneumothorax develops, respectively.

Carriers of the *FLCN* germline alteration can undergo abdominal cross-sectional imaging beginning at age 20, to be performed every 3 years [220,221,222,223,224]. The predominant renal tumor subtypes in BHD, HOCT and chromophobe, tend to be indolent [225], with median growth rate of 0.1 cm/year [63]. Therefore, active surveillance of renal masses is practiced, with intervention deferred until the dominant renal lesion reaches 3 cm. As of yet, no known metastasis has developed in patients with BHD when following this management strategy [63,226]. Furthermore, advancements in imaging techniques are attempting to differentiate benign oncocytomas and the more malignant chromophobe or HOCT lesions in an attempt to prevent unnecessary surgery [227].

## 9. Translocation RCC

### 9.1. Clinical Aspects of Translocation RCC

Translocation RCCs were first described in 1996 and are associated with abnormal gene fusion of the MiT (micro-opthalmia transcription) class of transcription factors, such as MiTF.; TFE3; and TFEB [228]. In 2011, a germline alteration in MiTF was first identified (p.E318K), with predisposition for both renal cancer and cutaneous melanoma [229,230]. In addition, translocation RCC is more prevalent in children and young adults although cases may present later in life [231,232]. Renal tumors can have histological features that overlap with clear cell or papillary RCC, with definite diagnosis provided by karyotype or immunostaining to note the translocation or fusion product amplification, respectively. The classification of translocation RCC continues to advance as TFE3 and TFEB altered RCC are not separately classified by the WHO as they behave very uniquely in that TFE3 is far more common and aggressive that TFEB. However, TFEB amplification can behave aggressively when compared to TFEB translocation [7].

### 9.2. Mechanism of Tumorigenesis of Translocation RCC

The MiT family of transcription factors are implicated in cell growth, differentiation, and metabolism, typically in response to nutrient availability [233]. Activity of these transcription factors are predicated on nuclear localization, with regulation by PTMs such as ubiquitination; acetylation; phosphorylation (via mTORC1, see BHD section); and SUMOylation (small ubiquitin like modifiers) [234]. Somatic translocation RCC is characterized by gene fusions (i.e., TFE3-PRCC t(X.;1)(p11.2;q21.1) that result in continued presence of transcription factor in the nucleus, with resultant upregulation of downstream targets contributing to oncogenesis [228,231,235]. In the case of the MITF p.E318K variant, loss of SUMOylation site results, with subsequent inappropriate activation of the transcription factor [229]. Furthermore, crosstalk between folliculin, mTOR and MiT family transcription factors likely contributes to shared tumorigenesis mechanisms between BHD, tuberous sclerosis and translocation RCC [114,215]. Upregulation of genes implicated in apoptosis inhibition (*BIRC7*); invasion (*ACP5*); and cell cycle (*CCND1)* occurs [230]. Additionally, melanocytic markers such as glycoprotein NMB (*GPNMB)* and *MLANA* have increased expression, accounting for the cutaneous melanoma phenotype seen with this hereditary syndrome.

### 9.3. MiTF Therapeutic Targets and Treatment Strategies

GPNMB is directly upregulated in MITF-driven tumors and may serve as a therapeutic target for patients with germline *MITF* variant that have advanced disease [212,230,236]. Indeed, an antibody drug conjugate against GPNMB has shown efficacy in patients with GPNMB expressing breast cancer and melanoma [237,238,239].

As this hereditary syndrome was only recently discovered, only limited clinical experience exists, and there is no consensus as to screening and management of renal tumors. Translocation RCCs, especially in adults, are associated with a more aggressive biology when compared to other types of sporadic RCC (e.g., more infiltrative in nature with advanced presentation) [240]. Thus, if active surveillance is employed for a renal mass in a patient with *MITF p.E318K* variation, the tumor should be closely followed.

## 10. Concluding Thoughts

Renal cell carcinoma is not a single entity, but rather represents a constellation of tumors defined by distinct clinical and genetic signatures, with several RCC subtypes having overlapping cell biology alterations which converge on alterations to proteins e.g., mTOR and HIF (see Table 1). Efforts at characterizing the mechanisms of tumorigenesis through which genetic alterations result in unregulated cell growth and/or dysfunctional metabolism have been pioneered via the study of patients with hereditary RCC (HRC). Furthermore, PTMs appear to play a major role in tumorigenesis through oncometabolite building, leading to changes in acetylation, succinylation, succination and sumoylation. The molecular changes of these germline alterations can be targeted with drug therapy, such as belzutifan for VHL, with the resultant novel application of systemic treatment as an option for management of localized kidney cancer, as opposed to its traditional relegation to only treating metastatic disease (see Table 2 highlighting clinical trials targeting subtype-specific RCC). Similar advances may be expected in other hereditary syndromes (i.e., BHD) as our understanding of the molecular pathogenesis of these entities continues to mature. Morbidity and mortality outcomes are expected to continue to improve as further elucidation of hereditary RCC tumorigenesis mechanisms evolves. Such elucidation has already led to improvements in surveillance protocols, surgical management (i.e., utilization of the 3 cm rule) and therapeutic options in an RCC subtype-specific fashion. As an example, prior to standardized surveillance protocols in VHL the average median survival was ~40 years old [89]. This survival has improved with regular whole-body imaging and refined surgical techniques, including the use of partial nephrectomy with the goal of avoiding hemodialysis and renal transplant. Novel agents, such as belzutifan, are expected to continue to improve the treatment and outcomes of hereditary RCC to minimize tumor development/growth, reduce the need for repeat surgery, delay the development of chronic kidney disease. This will lower hemodialysis rates, and lower rates of metastatic disease with an overall effect of attenuating morbidity and mortality. Furthermore, such agents may be used in the localized setting earlier in life prior to metastasis. In addition, future work will evaluate how such novel agents may be used with existing systemic therapies such as checkpoint inhibitors. As many of the disease pathways and genetic mutations of HRC are seen in sporadic kidney tumors (especially with VHL), it is hoped that we can both translate findings from the hereditary RCC population to the general population and continue to develop clinical trials for targeted drugs for both localized and advanced hereditary and/or sporadic renal tumors in order to move towards fully realizing precision medicine for all stages of kidney cancer.

## Figures and Tables

**Table 1 genes-13-02122-t001:** Genetic alterations, tumorigenesis mechanisms, clinical features and surgical management of renal lesions found in hereditary renal cell carcinoma syndromes.

Syndrome	Chromosome Localization	Gene(s)	Tumorigenesis Mechanisms:	Clinical Features	3-cm Rule Applied:
VHL	3p25	*VHL*	Augmented HIF Angiogenesis through VEGF lipid accumulation, β-oxidation inhibition aerobic glycolysis Augmented erythropoietin Mitochondrial lysine hyperacetylation	Brain, spine, retina and liver hemangioblastomas Pancreatic neuroendocrine tumors (NET) Endolymphatic sac tumors Clear cell renal cell carcinoma Pheochromocytoma/paragangliomas Benign cystadenomas of the epididymis or broad ligament	Yes
HPRC	7q31	*MET*	MET activation mutations or amplification Augmented cell growth, angiogenesis Increased motility Attenuated contact inhibition	Type 1 Papillary RCC (only finding in syndrome) Hypovascular/attenuated signal on CT/MRI	Yes
Succinate Dehydrogenase-Deficient Renal Cell Carcinoma	5p15.33-SDHA 1p36.13-SDHB 1q23.3-SDHC 11q23.1-SDHD 11q12.2-SDHAF2	*SDHA, SDHB, SDHC, SDHD, SDHAF2*	Defective oxidative phosphorylation with succinate acculmation (oncometabolite) Lysine succinylation of protein targets Elevated HIF1α	Pheochromocytomas Head/neck paragangliomas GIST tumors SDH-deficient RCC	No (wide local excision)
HLRCC	1q42.3-q43	*Fumurate hydratase*	Elevated fumurate (oncometabolite) Succination of cysteine residues on proteins Decreased mtDNA copy number Increased mtDNA mutations increased NRF2 activity (anti-oxidant system) Aerobic glycolysis	Cutaneous leiomyomas and uterine leiomyomas Hysterectomy at an early age HLRCC-associated RCC Histology: high grade tumors with large eosinophilic nucleoli and perinuclear clearing, increased S-(2-succinyl) cysteine staining (succination)	No (wide local excision)
BAP1	3p21	*BAP1*	Lost BAP1 deubiquitinase activity Augmented cell cycle progression (G1 to S) Inhibited DNA repair	Mesotheliomas of chest/abdomen Uveal melanoma Cutaneous melanoma Aggressive RCC	Unknown (likely not recommended)
Birt-Hogg-Dubé Syndrome (BHD)	17p11.2	*Folliculin*	Overactive AKT-mTOR TFEB overactivity (lost mTOR inhibition) Overative mTOR activity on p70S6K and 4E-BP	Benign painless cutaneous papules (fibrofolliculomas) Pulmonary cysts with risk for spontaneous pneumothorax Renal tumors:(oncocytoma, chromophobe, hybrid oncocytic) Possibly increased risk for colonic tumors	Yes
Tuberous Sclerosis	9q34 16p13	*TSC1, TSC2*	Overactive mTOR activity Augmented protein synthesis	Angiomyelolipomas RCC in ~2–4% Skin: angiofibromas, ash-leaf spots, Shagreen patches & periungual fibromas Brain: subependymal giant cell astrocytoma & cerebral cortical tubers Heart: rhabdomyoma Lung lymphangioleiomyomatosis (LAM) Epilepsy, behavioral disorder, and intellectual disabilities	Unknown for RCC AML: 4-cm rule
Translocation RCC	Xp11 t(6:11)	*TFE3, TFEB* also *MiTF/TFEC*	Elevated MiTF family activity Augmetned lysosomal biogenesis Increased autophagy Increased lysosomal exocytosis	Higher incidence in pediatric and young adults FISH, rtPCR and immunohistochemistry needed for diagnosis Histology: Xp11—increased GPNMB signal, papillary/nested ccRCC or cystic features t6:11- biphasic morphology	No (wide local excision)

Abbreviations: HIF—hypoxia inducible factor, VHL—von-Hippel Lindau, HPRC—hereditary papillary renal carcinoma, HLRCC—Hereditary leiomyomatosis and renal cell cancer, BAP1—BRCA1-associated protein 1, MET—mesenchymal-epithelial transition factor, SDH—succinate dehydrogenase, FISH—Fluorescence in situ hybridization, MITF—Microphthalmia-associated transcription factor, GIST—gastrointestinal stromal tumor, ccRCC—clear cell renal cell carcinoma, AML—angiomyelolipoma.

**Table 2 genes-13-02122-t002:** Highlighted clinical studies utilizing RCC subtype-specific therapies and their outcomes/limitations.

Hereditary RCC Subtype	Study Author or Trial Identifier	Agent(s) (Targeted pathway)	Study Phase	Dose/Duration	Outcomes/Trial Status	Limitations
VHL	Jonasch [69]	Belzutifan (HIF2α)	Phase II, open label, single group (n = 61)	120 mg daily	Median follow-up: 21.8 months Renal lesions: ORR 49% (CR or PR), stable disease 49% Pancreatic Leisons: ORR 79% (10% CR) CNS Hemangioblastomas: ORR 30% (6% CR) Retinal Hemangioblastomas: ORR 100% No grade 4 or 5 Aes	Low sample size No control arm
HPRC	Choueiri [94]	Foratenib (MET tyrosine kinase)	Phase II (n = 74)	Cohort A: 240 mg daily on days 1-5 every 14 days (intermittent arm) Cohort B: 80 mg daily	Patients risk stratified on basis of MET alteration (germline/somatic mutation, amplification, gain of chromosome 7) ORR: 13.5% (all patients) vs. 50% (germline mutation patients)MFS: 9.3 months	Low sample size No control arm
SDH-RCC	NCT03635892 NCT04068831	Cabozantinib/ Nivolumab Talzoparib/ Avelumab	Phase II (n = 97 goal) Phase II (n = 44 goal)	40 mg (Cabozantinib)/240 mg (Nivolumab) until disease progresses 1 mg (Talzoparib)/800 mg Avelumab	Study status: recruiting for both trials	Results pending
HLRCC	Srinivasan [147]	Bevacizumab/ Erlotinib (HIF/angiogenesis)	Phase II (n = 41)	Bevacizumab: 10 mg q2 weeks Erlotinib 150 mg PO daily until disease progression or unacceptable toxicity	ORR: 35% for sporadic type II papillary RCC vs. 72% for HLRCC PFS: 8.7 months (sporadic) vs. 21.1 months (HLRCC)	Low sample size No control arm
TSC	Bissler [194]	Everolimus (mTOR)	Phase III, RCT, double-blinded (n = 112)	10 mg daily	ORR: 54% for AML with median treatment length of 28.9 months ≥30% tumor volume reduction: 81.6% by week 96 ≥50%: 64.5% (49/76) by Week 96	RCC lesions not included in study

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
