# Peer review of "Tumorigenesis Mechanisms Found in Hereditary Renal Cell Carcinoma: A Review"

_genes, 2022, doi:10.3390/genes13112122_

Round 1

Reviewer 1 Report

In this review article, Webster BR et al have summarized all the important findings about the underlying mechanisms, which govern initiation and progression of renal cell carcinoma. They also highlight the new therapeutic targets that can be exploited for improving outcomes in patients with this debilitating disease. Overall, the review is very comprehensive and the authors have done a good job of summarizing important findings in the field. It will be very helpful if the authors could include a schematic figure/ a diagram that summarizes the key findings or provides a snapshot of the article, which will be extremely useful to the reader.

Author Response

- tables 1 and 2 have been added highlighting the types of tumorigenesis mechanisms for each RCC subtype and overview of key clinical trials highlighted in the paper, respectively 

Reviewer 2 Report

Thank you for the opportunity to review the manuscript entitled “Tumorigenesis Mechanisms Found in Hereditary Renal Cell Carcinoma: A Review”. The clinical topic is e important. However, I have several comments to improve the quality of the manuscript.

  1. It would benefit the paper if the authors included some overview tables for the studies included in the review. Could the authors mention pros as cons (limitations) to the final studies included in an overview table?
  2. Could the authors provide citations for lines 16-18?
  3.  Regarding lines 19-21, it is important to note that increasing evidence also mention that genetics and environmental factors (e.g., drinking water pollution) are likely to play a role. This could be incorporated.
  4. Could the authors provide a reference for lines 30-33?
  5. It would benefit the paper if the authors in the abstract went over what they will discuss in the paper already in the abstract, e.g., VHL, HPRC, TSC, Birt-Hogg-Dubé, Translocation RCC. Moreover, as mentioned above, if the authors could provide overview tables for included papers for each subsection that would benefit the paper.
  6. The author team references a Table 1, but I do not see it. Is it possible that it was not submitted together with the manuscript?
  7. I am missing a discussion on future implications. E.g., as science evolves how will important endpoints such as mortality and morbidity, etc. possibly change?
  8. A diagram that gives an overview of the different hereditary RCCs would be great. This could include typical clinical presentations or other relevant information.

Author Response

1) Table 2 has been added to highlight the clinical trials discussed in the paper

2) This has been cited.

3) A discussion of environmental risk factors (including asbestos and arsenic) is now in the introduction.

4) Lines 30-33 have been cited. 

5) The abstract now includes the conditions the review discusses and tables 1 and 2 are now included highlighting each subsection.

6) Table 1 is now uploaded.

7) The 'concluding thoughts' section now includes a discussion on how clinical outcomes are expected to improve with future use of novel agents.

8) Table 1 gives an overview of each RCC subtype. 

Round 2

Reviewer 2 Report

Great work